# Refinement Asymptotic Formulas of Eigenvalues and Eigenfunctions of a Fourth Order Linear Differential Operator with Transmission Condition and Discontinuous Weight Function

**Rando Rasul Qadir** *  **and Karwan Hama Faraj Jwamer**

Department of Mathematics, College of Basic Education, University of Sulaimani, P.O. Box 46, Sulaimani, Kurdistan Region of Iraq
* Correspondence: rando.qadir@univsul.edu.iq; Tel.: +96-4750-2086-654

**Abstract:** In this paper, we promote the refinement method for estimating asymptotic expression of the fundamental solutions of a fourth order linear differential equation with discontinuous weight function and transmission conditions. These refinement solutions utilize more accurate asymptotic formulas for the eigenvalues and eigenfunctions for the problem.

**Keywords:** differential operator; spectral parameter; fundamental solution; transmission condition; Krein space; dense set

---

## 1. Introduction

Differential equations with boundary conditions, especially Sturm-Liouville with a spectrum contained in the boundary condition, have various applications for mathematical physics, economics and biophysics. For instance, the vibration of strings and mass transfer [1–4]. The method of finding the eigenvalues and eigenfunctions of an eigenvalue problem was investigated by many authors. Naimark [5] studied a general linear differential operator of $n$th order. He obtained an asymptotic formula for the fundamental solutions, eigenvalues and eigenfunctions for the problem. Eventually, Kerimov and Mamedov [6] investigated a second order differential operator. They obtained an accurate asymptotic formulas for eigenvalues and eigenfunctions compared to Naimark's work.

The method of obtaining a refinement fundamental solution of a linear differential operator was studied in Reference [5] (Section 4.6). In this method, the author considered that the coefficients of the differential operator and their derivatives up to order $m$ are continuous. This method leads to a more accurate asymptotic formulas for eigenvalues and eigenfunctions, indeed. For more detail about the method we recommend the reader to see Reference [5] (Section 4.6).

More and more authors have been interested in investigating this kind of subject in recent years. Fourth order linear differential equations have extensive applications in different fields of engineering and science. For instance, several appropriate mathematical models have been suggested in References [3,4] which assist in describing the oscillation behavior appearing in the actual suspension bridges. Nowadays, investigating the eigenvalue problem is going in the direction of discontinuity of the solutions or the coefficients of the differential operator with transmission conditions at the point of discontinuities. Jwamer [7] studied the asymptotic behavior of the eigenvalues and eigenfunction for a second order differential equation, where its cofficients are discontinuous. Recently, Bayramoglu, Bayramov and Şen, in References [2,8–10] investigated second order and fourth order differential operators with a spectral parameter contained in the boundary conditions and transmission conditions.

Şen formulated a new method for determining linearly independent solutions, eigenvalues and eigenfunctions for the eigenvalue problems.

Concerning Reference [2], we study a fourth order differential operator with the eigenparameter contained in the boundary conditions and transmission conditions. In the present paper, we use the refined method for obtaining the linearly independent solution of the problem. Moreover, an accurate asymptotic formula was obtained for eigenvalues and eigenfunctions. First, we constitute the problem as follows:

Consider a fourth order differential operator of the form:

$$L[y] = y^{(4)}(x) + p(x)y' + q(x)y(x) = \lambda\omega(x)y(x) \tag{1}$$

where, $x$ lies in the interval $I = [-1,0) \cup (0,1]$. We assume that all solutions of (1) are in $L^2(I)$ and they are satisfying the following boundary conditions at the boundary points $x = -1$ and $x = 1$

$$L_1[y] = \alpha_1 y(-1) - \alpha_2 y'''(-1) = 0 \tag{2}$$

$$L_2[y] = \beta_1 y'(-1) - \beta_2 y''(-1) = 0 \tag{3}$$

$$L_3[y] = \lambda y(1) - y'''(1) = 0 \tag{4}$$

$$L_4[y] = \lambda y'(1) - y''(1) = 0 \tag{5}$$

At the discontinuity point $x = 0$, the solutions of Equation (1) satisfies the following transmission conditions:

$$L_5(y) = y(0+) - y(0-) = 0 \tag{6}$$

$$L_6(y) = y'(0+) - y'(0-) = 0 \tag{7}$$

$$L_7(y) = y''(0+) - y''(0-) + \lambda\delta_1 y'(0-) = 0 \tag{8}$$

$$L_8(y) = y'''(0+) - y'''(0-) + \lambda\delta_2 y(0-) = 0 \tag{9}$$

where, the weight function $\omega(x)$ defined on the interval $I$ as follows:

$$\omega(x) = \begin{cases} \omega_1^4; & x \in [-1,0) \\ \omega_2^4; & x \in (0,1] \end{cases} \tag{10}$$

The functions $p(x)$, $q(x)$ are continuous on the interval $I$, $p(x) \in C^2[-1,1]$ and they have finite limits, $p(0\pm) = \lim_{x \to 0\pm} p(x)$, $q(0\pm) = \lim_{x \to 0\pm} q(x)$. $\lambda$ is the spectral parameter, $\alpha_i, \beta_i$ and $\omega_i$ are real scalars (for $i = 1, 2$), with $|\omega_1 + \omega_2| \neq 0$ and $\int_{-1}^{1} p(x)dx \neq 0$.

In Section 2, we recall some definitions and theorems that are useful in the next section. Also, we formulate the problem and investigate its properties in operator theory views. The most important result is the estimation of the eigenvalues and eigenfunctions (see Theorem 4). According to Theorem 3, the eigenvalue problem (1)–(9) possesses infinitely many positive and negative eigenvalues. From Theorem 4, we obtain that the zeros of Equations (21) and (22) are the eigenvalues of the problem. We assume that the solutions of (1) satisfy the initial conditions (15)–(19). The purpose of these conditions is to conduct a more accurate system of solution of Equation (1), such that it satisfies the boundary and transmission conditions. In Section 3, we determine the refinement asymptotic formulas for the linearly independent solutions (resp. their derivatives) of Equation (1). Every solution satisfies the initial conditions (15)–(19), respectively. Also, we estimate an upper bound for the fundamental solutions. In Section 4, we establish the accurate asymptotic formulas for the eigenvalues and eigenfunctions of the problem (1)–(9). Finally, we show that the problem in Reference [2] is a special case of the problem that is presented in this paper.

## 2. Preliminaries and Constructions

In this section, we begin with the following definitions and theorems, which are necessary in the next section. The definitions and notations in this paper can be found in Reference [1]. Also, we construct the initial conditions for which the linearly independent solution of the problem will satisfy:

**Definition 1** ([2]). *Suppose that $f, g \in L^2(I)$, then the inner product of $f, g$ on $L^2(I)$ defined as follows:*

$$\langle f, g \rangle_1 = \int_{-1}^{0} f_1 \bar{g}_1 dx + \int_{0}^{1} \bar{f}_2 g_2 dx \tag{11}$$

*where, $f_1 = f(x)|_{[-1,0)}$, $f_2 = f(x)|_{(0,1]}$ and $\bar{f}_2$ is the complex conjugate of $f_2$. It is provided that $(L^2(I), \langle \cdot \rangle)_1$ form a Hilbert space.*

**Definition 2** ([2]). *Let $\mathbb{K} = L^2(I) \oplus \mathbb{C} \oplus \mathbb{C} \oplus \mathbb{C}_{\delta_1} \oplus \mathbb{C}_{\delta_2}$. Then define $[.]$ on $\mathbb{K}$ as follows:*

$$[F, G] = \langle f, g \rangle_1 + \langle h_1, k_1 \rangle + \langle h_2, k_2 \rangle + \langle h_3, k_3 \rangle + \langle h_4, k_4 \rangle \tag{12}$$

*where, $F = (f, h_1, h_2, h_3, h_4)$ and $G = (g, k_1, k_2, k_3, k_4)$ are in $\mathbb{K}$. Then $\mathbb{K}$, under the inner product $[.]$ form a direct sum of modified Krein spaces.*

Note that Krein space is a linear space $H$ with an inner product $[.,.]$ defined on $H$ such that there exists a decomposition $H = H_+ \oplus H_-$, where $(H_\pm, \pm[.,.])$ are Hilbert spaces and $[H_+, H_-] = 0$ [1]. The elements of a Krein space $H$ are classified in terms of the inner product $[.,.]$ as follows: an element $x$ in $H$ is called positive (negative), if $[x, x] > 0$ ($[x, x] < 0$). A linear manifold or a subspace $K \subseteq H$ is called positive (negative), if all its non-zero elements are positive (negative). If $f \in H$, then there exists $f_\pm \in H_\pm$, such that $f = f_+ + f_-$. Also, a projection $P_\pm$ can be defined on $H$ as follows: $P_\pm f = f_\pm$. Then the inner product $(.,.)$ defined in terms of $[.,.]$ with the operator $J = P_+ - P_-$ by:

$$(f, g) = [Jf, g]$$

The operator $J$ is said to be a fundamental symmetry of the Krein space $H$. The reader can see References [1,11] for more detail about Krein spaces. Now, we turn to finding a formula for estimating the eigenvalues of the problem (1)–(5). This method is mentioned in Reference [2]: Suppose that $J_0 : L^2(I) \to L^2(I)$ is introduced by $(J_0 f)(x) = f(x)$. Then consider a fundamental symmetry operator $J$ on the Krein space $\mathbb{K}$

$$J = \begin{bmatrix} J_0 & 0 & 0 & 0 & 0 \\ 0 & 1 & 0 & 0 & 0 \\ 0 & 0 & 1 & 0 & 0 \\ 0 & 0 & 0 & sgn\delta_1 & 0 \\ 0 & 0 & 0 & 0 & sgn\delta_2 \end{bmatrix}$$

where, $sgnx$ refers to the signum function of a real number $x$ and it is defined as follows

$$sgnx = \begin{cases} 1: & x > 0 \\ 0: & x = 0 \\ -1: & x < 0 \end{cases} \tag{13}$$

It is obvious that $\langle .,. \rangle = [J.,.]$ form a positive definite inner product on $\mathbb{K}$. This implies that $\mathbb{K}$ is a Hilbert space with the inner product $\langle .,. \rangle = [J.,.]$ defined by $\mathbb{K}_0 := (\mathbb{K}, \langle .,. \rangle)$. Let $A$ be a linear operator defined with respect to the conditions of our problem (1)–(9):

$$D(A) := \{(f, h_1, h_2, h_3, h_4) \in \mathbb{K} | f^{(i)} \in AC_{loc}((0,1)), i = \overline{1,3}, Lf \in L^2(I), L_k = 0,$$
$$k = \overline{1,4}, h_1 = f(1), h_2 = f'(1), h_3 = -\delta_1 f'(0), h_4 = -\delta_2 f(0)\}$$

Then for any $F = (f, f(1), f'(1), -\delta_1 f'(0), -\delta_2 f(0)) \in D(A)$, we have

$$AF = (Lf, -f'''(1), -f''(1), f''(0+) - f''(0-), f'''(0+) - f'''(0-)),$$

An interesting application of this process is to rewriting the problem (1)–(9) in the form of operators as follows:

$$AF = \lambda F$$

Consider $X$ as a normed linear space. A subset $G$ of $X$ is said to be dense in $X$, if every element $x$ of $G$ is the limit point for a sequence in $X$. In the following theorem, we investigate the density of the domain of the operator $A$, which constructed in term of the problem (1)–(9) in $\mathbb{K}_0$. Consequently, we show that the operator $A$ is self-adjoint in $\mathbb{K}_0$.

**Theorem 1.** *The set $D(A)$, the domain of the operator $A$, is a dense set in the Hilbert space $\mathbb{K}_0$.*

**Proof.** Suppose that $\phi(x)$ is a function defined by:

$$\phi(x) = \begin{cases} \phi_1(x): & x \in [-1,0) \\ \phi_2(x): & x \in (0,1] \end{cases} \tag{14}$$

where, $\phi_1(x) \in C^\infty(-1,0)$ and $\phi_2(x) \in C^\infty(0,1)$. We denote $W$ as the set of all functions of the form $\phi(x)$. It is easy to see that, $W$ is dense in $\mathbb{K}_0$, (see Reference [12] (Lemma 2.1)). Hence $D(A)$ is dense set in $\mathbb{K}_0$. $\square$

**Theorem 2.** *The operator $A$ is self-adjoint in $\mathbb{K}_0$.*

**Proof.** The proof is a direct consequence of [12] (Theorem 2.2). $\square$

This leads us to deal with the problem in the direction of operator theory. We can compare the properties of this operator at a higher level in the Krein or Hilbert spaces. The most important result of this construction is the eigenvalues and eigenfunctions of the problem, (1)–(9) are the eigen values and eigenvectors of the operator $A$. The following theorem guarantees that the set of eigenvalues and eigenfunctions of the problem (1)–(9) is not empty:

**Theorem 3.** *The operator $A$ has infinity many positive (negative) eigenvalues. Moreover, every eigenvalue has a corresponding eigenfunction.*

**Proof.** It is obvious by Reference [11] (Proposition 1.8). $\square$

**Lemma 1.** *Suppose that $p(x)$, $q(x)$ are continuous functions on the interval $[-1,1]$, $\omega(x)$ is a weight function defined as in (10) and $C_i(\lambda)$, (for $i = \overline{1,4}$) are entire on $\mathbb{C}$. Then for any arbitrary complex number $\lambda$ and $a \in I$ the differential equation*

$$y^{(4)}(x) + p(x)y' + q(x)y(x) = \lambda\omega(x)y(x)$$

*has a unique solution, for which satisfy the initial conditions*

$$y(a) = C_1(\lambda), y'(a) = C_2(\lambda), y''(a) = C_3(\lambda), y'''(a) = C_4(\lambda)$$

**Proof.** Since the functions $p(x)$ and $q(x)$ are continuous on the interval $[-1, 1]$, then by using the Existence and Uniqueness Theorem in the theory of differential equations. The proof is hold directly. □

If $a = -1$, then by Lemma 1 there exists a solution $u_{11}(x, \lambda)$ of the differential Equation (1) on the interval $[-1, 0)$, such that it satisfies the initial conditions,

$$u_{11}(-1) = \alpha_2, u'_{11}(-1) = 0, u''_{11}(-1) = 0 \ u'''_{11}(-1) = -\alpha_1 \tag{15}$$

According to this solution, we choose another solution $u_{12}(x, \lambda)$ of the differential Equation (1) on the interval $(0, 1]$, such that at $a = 0$ satisfy the initial conditions,

$$u_{12}(0) = u_{11}(0), u'_{12}(0) = u'_{11}(0), u''_{12}(0) = u''_{11}(0) - \lambda \delta_1 u'_{11}(0)$$
$$and \ u'''_{12}(0) = u'''_{11}(0) - \lambda \delta_2 u_{11}(0) \tag{16}$$

By the same way, use Lemma 1, we can define two solutions $u_{21}(x, \lambda)$ and $u_{22}(x, \lambda)$ on the intervals $[-1, 0)$ and $(0, 1]$, respectively. These solutions satisfy the following initial conditions:

$$\begin{cases} u_{21}(-1) = 0, u'_{21}(-1) = \beta_2, u''_{21}(-1) = -\beta_1 \ u'''_{21}(-1) = 0 \\ u_{22}(0) = u_{21}(0), u'_{22}(0) = u'_{21}(0), u''_{22}(0) = u''_{21}(0) - \lambda \delta_1 u'_{21}(0) \\ and \ u'''_{22}(0) = u'''_{21}(0) - \lambda \delta_2 u_{21}(0) \end{cases} \tag{17}$$

Dually, for the Equations (15) and (16), we can choose $v_{11}(x, \lambda)$ and $v_{12}(x, \lambda)$ as two solutions of Equation (1), such that they are satisfying the following initial conditions on the intervals $[-1, 0)$ and $(0, 1]$, respectively:

$$\begin{cases} v_{12}(1) = -1, v'_{12}(1) = 0, v''_{12}(1) = 0 \ v'''_{12}(1) = \lambda \\ v_{11}(0) = v_{12}(0), v'_{11}(0) = v'_{12}(0), v''_{11}(0) = v''_{12}(0) + \lambda \delta_1 v'_{12}(0) \\ and \ v'''_{11}(0) = v'''_{12}(0) + \lambda \delta_2 v_{12}(0) \end{cases} \tag{18}$$

Finally, we have to find two more solutions, to compute the formula for the eigenvalues and eigenfunctions of the problem (1)–(9). Again by using Lemma 1, Equation (1) possess two solutions $v_{21}(x, \lambda)$ and $v_{22}(x, \lambda)$ on the intervals $[-1, 0)$ and $(0, 1]$, respectively. Such that they are satisfying the conditions:

$$\begin{cases} v_{22}(1) = 0, v'_{22}(1) = -1, v''_{22}(1) = \lambda \ v'''_{22}(1) = 0 \\ v_{21}(0) = v_{22}(0), v'_{21}(0) = v'_{22}(0), v''_{21}(0) = v''_{22}(0) + \lambda \delta_1 v'_{22}(0) \\ and \ v'''_{21}(0) = v'''_{22}(0) + \lambda \delta_2 v_{22}(0) \end{cases} \tag{19}$$

From Theorem 3, we conclude that problem (1)–(9) has infinitely many positive and negative eigenvalues. Now, we turn to the questions "What are the asymptotic behavior of the eigenvalues? How can we determine them?" Consider a general differential operator of the form:

$$L[y] = y^{(n)}(x) + p_{n-2}(x)y^{(n-1)}(x) + \cdots + y = \lambda \omega(x)y(x)$$

The Existence Theorem [13] conducts that the above differential equation has a fundumental system of $n$ linearly independent solutions $y_1, y_2, \ldots, y_n$ on the interval $[a, b] \subseteq \mathbb{R}$. If this differential equation is constrained with the boundary conditions $L_i[y]_{a,b} = \alpha_i$, for $i = 1, \ldots, n$, then the eigenvalues of the eigenvalue problem are zeros of the equation:

$$
\begin{vmatrix}
L_1[y_1] & L_1[y_2] & \dots & L_1[y_n] \\
L_2[y_1] & L_2[y_2] & \dots & L_2[y_n] \\
\vdots & \vdots & \vdots & \vdots \\
L_n[y_1] & L_n[y_2] & \dots & L_n[y_n]
\end{vmatrix} = 0
\tag{20}
$$

Substituting the boundary conditions (2)–(5) in the Equation (20), we obtain the following Wronskians

$$
W_1(\lambda) =
\begin{vmatrix}
u_{11}(x,\lambda) & u_{21}(x,\lambda) & v_{11}(x,\lambda) & v_{21}(x,\lambda) \\
u_{11}'(x,\lambda) & u_{21}'(x,\lambda) & v_{11}'(x,\lambda) & v_{21}'(x,\lambda) \\
u_{11}''(x,\lambda) & u_{21}''(x,\lambda) & v_{11}''(x,\lambda) & v_{21}''(x,\lambda) \\
u_{11}'''(x,\lambda) & u_{21}'''(x,\lambda) & v_{11}'''(x,\lambda) & v_{21}'''(x,\lambda)
\end{vmatrix}
\tag{21}
$$

and

$$
W_2(\lambda) =
\begin{vmatrix}
u_{12}(x,\lambda) & u_{22}(x,\lambda) & v_{12}(x,\lambda) & v_{22}(x,\lambda) \\
u_{12}'(x,\lambda) & u_{22}'(x,\lambda) & v_{12}'(x,\lambda) & v_{22}'(x,\lambda) \\
u_{12}''(x,\lambda) & u_{22}''(x,\lambda) & v_{12}''(x,\lambda) & v_{22}''(x,\lambda) \\
u_{12}'''(x,\lambda) & u_{22}'''(x,\lambda) & v_{12}'''(x,\lambda) & v_{22}'''(x,\lambda)
\end{vmatrix}
\tag{22}
$$

After a simple calculation, we conclude that the Wronskians are equivalent. Means that $W_1(\lambda) = W_2(\lambda)$, for any values of the spectral parameter $\lambda \in \mathbb{C}$. Hence, from (20)–(22), we have the following interesting result for deriving a formula to estimate the eigenvalues of the problem in the interval $I$:

**Theorem 4.** *Consider the problem (1)–(9), then the eigenvalues are the zeros of the Wronskian $W_1(\lambda) = W_2(\lambda)$.*

**Proof.** It is similar to the proof of [2] (Theorem 3.2). □

## 3. Asymptotic Behavior of The fundamental Solutions

In this section, we establish the refinement asymptotic formulas for the linearly independent solutions (resp. their derivatives) of Equation (1), such that each solution satisfy the initial conditions (15)–(19), respectively. Also, we estimate an upper bound for the solution.

**Lemma 2.** *If $\lambda = s^4$, $s = \sigma + i\tau$ and $k = 0, 1, 2, 3$. Then the fundamental solutions (15)–(17) satisfy the following asymptotic formulas:*

$$
\frac{d^k}{dx^k} u_{11}(x,\lambda) = \frac{d^k}{dx^k} \left( \frac{\alpha_2}{2} + \frac{\omega_1^2 \alpha_2}{8s^2} \int_{-1}^{x} p(t)dt \right) \cos(s\omega_1(x+1))
$$

$$
+ \frac{d^k}{dx^k} \left( \frac{\alpha_2}{2} - \frac{\omega_1^2 \alpha_2}{8s^2} \int_{-1}^{x} p(t)dt \right) \cosh(s\omega_1(x+1))
$$

$$
+ O\left( |s|^{k-3} e^{|s\omega_1|(x+1)} \right)
\tag{23}
$$

$$
\frac{d^k}{dx^k} u_{12}(x,\lambda) = \frac{d^k}{dx^k} \left( \frac{(s\omega_2)^2 \delta_1 u_{11}'(0)}{2} + \frac{\omega_2^4 \delta_1 u_{11}'(0)}{8} \int_{0}^{x} p(t)dt \right) \cos(s\omega_2 x)
$$

$$
+ \frac{s\omega_2 \delta_1 u_{11}(0)}{2} \frac{d^k}{dx^k} \sin(s\omega_2 x) - \frac{(s\omega_2)^2 \delta_1 u_{11}'(0)}{2} \frac{d^k}{dx^k} \cosh(s\omega_2 x)
$$

$$
- \frac{d^k}{dx^k} \left( \frac{(s\omega_2) \delta_2 u_{11}(0)}{2} - \frac{\omega_2^4 \delta_1 u_{11}'(0)}{8} \int_{0}^{x} p(t)dt \right) \sinh(s\omega_2 x)
$$

$$
+ O\left( |s|^{k} e^{|s\omega_2|(x+1)} \right)
\tag{24}
$$

$$
\begin{aligned}
\frac{d^k}{dx^k} u_{21}(x,\lambda) = \quad & \frac{\beta_2}{2(s\omega_1)^2} \frac{d^k}{dx^k} \cos\left(s\omega_1(x+1)\right) \\
& + \frac{d^k}{dx^k}\left(\frac{\beta_2}{2s\omega_1} + \frac{\omega_1\beta_2}{8s^3}\int_{-1}^{x} p(t)dt\right)\sin\left(s\omega_1(x+1)\right) \\
& + \frac{\beta_1}{2(s\omega_1)^2}\frac{d^k}{dx^k}\sinh\left(s\omega_1(x+1)\right) \\
& - \frac{d^k}{dx^k}\left(\frac{\beta_2}{2(s\omega_1)} + \frac{\omega_1\beta_2}{8s^3}\int_{-1}^{x}p(t)dt\right)\cosh\left(s\omega_1(x+1)\right) \\
& + O\left(|s|^{k-4}e^{|s\omega_1|(x+1)}\right)
\end{aligned}
\tag{25}
$$

$$
\begin{aligned}
\frac{d^k}{dx^k}u_{22}(x,\lambda) = \quad & \frac{(s\omega_2)^2\delta_1 u_{21}'(0)}{2}\frac{d^k}{dx^k}\cos(s\omega_2 x) + \frac{s\omega_2\delta_1 u_{21}'(0)}{2}\frac{d^k}{dx^k}\sin(s\omega_2 x) \\
& - \frac{(s\omega_2)^2\delta_1 u_{21}'(0)}{2}\frac{d^k}{dx^k}\cosh(s\omega_2 x) \\
& - \frac{d^k}{dx^k}\left(\frac{(s\omega_2)\delta_2 u_{21}(0)}{2} - \frac{\omega_2^4\delta_1 u_{21}'(0)}{8}\int_0^x p(t)dt\right)\sinh(s\omega_2 x) \\
& + O\left(|s|^{k-1}e^{|s\omega_2|(x+1)}\right)
\end{aligned}
\tag{26}
$$

**Proof.** Consider the Differential Equation (1), then we can rewrite is as follows:

$$
y^{(4)}(x) - s^4\omega(x)y(x) = m(x)
\tag{27}
$$

where, $m(x) = -p(x)y'(x) - q(x)y(x)$, then Equation (27) has a unique linearly independent solution $u_{11}(x,\lambda)$ on $[-1,0]$, which satisfy the initial condition (15) by Lemma 1. It is easy to show that, $e^{s\omega_1 x}$, $e^{-s\omega_1 x}$, $e^{is\omega_1 x}$ and $e^{-is\omega_1 x}$ are the linearly independent solutions of the equation:

$$
y^{(4)}(x) - s^4\omega_1^4 y(x) = 0
$$

By using the method Variation of parameters, we can see the solution $u_{11}(x,\lambda)$ has the form:

$$
\begin{aligned}
u_{11}(x,\lambda) = \quad & \frac{\alpha_2}{2}\cos\left(s\omega_1(x+1)\right) + \frac{\alpha_1}{(2s\omega_1)^3}\sin\left(s\omega_1(x+1)\right) \\
& + \left(\frac{\alpha_2}{4} - \frac{\alpha_1}{(2s\omega_1)^3}\right)e^{(s\omega_1(x+1))} + \left(\frac{\alpha_2}{4} + \frac{\alpha_1}{(2s\omega_1)^3}\right)e^{(-s\omega_1(x+1))} \\
& - \frac{\omega_1}{4s^3}\int_{-1}^{x}\left[-2\sin\left(s\omega_1(x-t)\right) + e^{(s\omega_1(x-t))} - e^{(-s\omega_1(x-t))}\right]p(t)u_{11}'(t)dt \\
& - \frac{\omega_1}{4s^3}\int_{-1}^{x}\left[-2\sin\left(s\omega_1(x-t)\right) + e^{(s\omega_1(x-t))} - e^{(-s\omega_1(x-t))}\right]q(t)u_{11}(t)dt
\end{aligned}
\tag{28}
$$

Now, we have to calculate the first integral part of Equation (28), this requires $u_{11}'(x,\lambda)$, by differentiating Equation (28), we obtain

$$u'_{11}(x,\lambda) = \quad -(s\omega_1)\frac{\alpha_2}{2}\sin(s\omega_1(x+1)) + (s\omega_1)\frac{\alpha_1}{(2s\omega_1)^3}\cos(s\omega_1(x+1))$$

$$+(s\omega_1)\left(\frac{\alpha_2}{4} - \frac{\alpha_1}{(2s\omega_1)^3}\right)e^{(s\omega_1(x+1))} - (s\omega_1)\left(\frac{\alpha_2}{4} + \frac{\alpha_1}{(2s\omega_1)^3}\right)$$

$$\times e^{(-s\omega_1(x+1))}$$

$$-\frac{\omega_1^2}{4s^2}\int_{-1}^{x}\left[-2\cos(s\omega_1(x-t)) + e^{(s\omega_1(x-t))} + e^{(-s\omega_1(x-t))}\right]p(t)u'_{11}(t)dt$$

$$-\frac{\omega_1^2}{4s^2}\int_{-1}^{x}\left[-2\cos(s\omega_1(x-t)) + e^{(s\omega_1(x-t))} + e^{(-s\omega_1(x-t))}\right]q(t)u_{11}(t)dt \quad (29)$$

If we set $F'_{11}(x,s) = (s\omega_1)^{-1}e^{|s\omega_1|(x+1)}u'_{11}(x,s)$, $M' = \max_{x\in[-1,0)}|F'_{11}(x,s)|$, $F_{11}(x,s) = e^{|s\omega_1|(x+1)}u_{11}(x,s)$ and $M = \max_{x\in[-1,0)}|F_{11}(x,s)|$, then after a simple calculation we obtain:

$$|F'_{11}(x,s)| \leq \frac{|\alpha_2|}{4} + \frac{\omega_1}{4s^2}M'\int_{-1}^{x}|p(t)|dt + \frac{\omega_1}{4s^3}M\int_{-1}^{x}|q(t)|dt$$

Since the functions $p(x)$ and $q(x)$ are real valued continuous functions on the interval $I$, then their integrals are bounded on $[-1,0)$. So, for a sufficiently large $|\lambda|$, we have $F'_{11}(x,s) = O(1)$. Thus, $u'_{11}(x,s) = O\left(|s|e^{|s\omega_1|(x+1)}\right)$ and $u_{11}(x,s) = O\left(e^{|s\omega_1|(x+1)}\right)$. Substituting these into Equation (29) and we get

$$u'_{11}(x,\lambda) = \quad -(s\omega_1)\frac{\alpha_2}{2}\sin(s\omega_1(x+1)) + (s\omega_1)\frac{\alpha_2}{4}e^{(s\omega_1(x+1))} - (s\omega_1)\frac{\alpha_2}{4}e^{(-s\omega_1(x+1))}$$

$$+O\left(|s|^{-2}e^{|s\omega_1|(x+1)}\right) \quad (30)$$

Putting Equation (30) into the first integral part of (28) and using the asymptotics of $u'_{11}(x,s)$, $u_{11}(x,s)$ and differentiating $k$ times with respect to $x$, then (23) is estimated. By using the same technique we obtain all the solutions (24)–(26). $\square$

**Lemma 3.** *If $\lambda = s^4$, $s = \sigma + i\tau$ and $k = 0,1,2,3$. Then the fundamental solutions (18)–(19) are satisfying the following asymptotic formulas:*

$$\frac{d^k}{dx^k}v_{12}(x,\lambda) = \quad \frac{-1}{2}\cos(s\omega_1(x-1)) - \left(\frac{s}{2\omega_2^3} + \frac{1}{8\omega_2 s}\int_x^1 p(t)dt\right)\sin(s\omega_2(x-1))$$

$$+\frac{s}{2\omega_2^3}\sinh(s\omega_2(x-1)) - \left(\frac{1}{2} - \frac{1}{8\omega_2 s}\int_x^1 p(t)dt\right)$$

$$\times\cosh(s\omega_2(x-1)) + O\left(|s|^{k-2}e^{|s\omega_2|(1-x)}\right) \quad (31)$$

$$\frac{d^k}{dx^k}v_{11}(x,\lambda) = \quad -\frac{d^k}{dx^k}\left(\frac{s^2\delta_1 v_{12}(0)}{2\omega_1^2} - \frac{\delta_1 v'_{12}(0)}{8}\int_x^0 p(t)dt\right)\cos(s\omega_1 x)$$

$$-\frac{s\delta_2 v_{12}(0)}{2\omega_1^3}\frac{d^k}{dx^k}\sin(s\omega_1 x) + \frac{s\delta_2 v_{12}(0)}{2\omega_1^3}\frac{d^k}{dx^k}\sinh(s\omega_1 x)$$

$$+\frac{d^k}{dx^k}\left(\frac{s^2\delta_2 v'_{12}(0)}{2\omega_1^2} - \frac{\delta_1 v'_{12}(0)}{8}\int_x^0 p(t)dt\right)\cosh(s\omega_1 x)$$

$$+O\left(|s|^{k+1}e^{|s\omega_1|(1-x)}\right) \quad (32)$$

$$\frac{d^k}{dx^k} v_{22}(x, \lambda) = \frac{d^k}{dx^k} \left( \frac{-s^2}{2\omega_2^2} - \frac{1}{2\omega_2} \int_x^1 p(t)dt \right) \cos\left(s\omega_2(x-1)\right)$$

$$+ \frac{d^k}{dx^k} \left( \frac{s^2}{2\omega_2^2} - \frac{1}{2\omega_2} \int_x^1 p(t)dt \right) \cosh\left(s\omega_2(x-1)\right)$$

$$+ O\left( |s|^{k+1} e^{|s\omega_2|(1-x)} \right) \tag{33}$$

$$\frac{d^k}{dx^k} v_{21}(x, \lambda) = -\frac{d^k}{dx^k} \left( \frac{s^2 \delta_1 v_{22}'(0)}{2\omega_1^2} + \frac{\delta_1 v_{22}'(0)}{2} \int_x^0 p(t)dt \right) \cos\left(s\omega_1 x\right)$$

$$- \frac{s\delta_2 v_{22}(0)}{2\omega_1^3} \frac{d^k}{dx^k} \sin\left(s\omega_1 x\right) + \frac{s\delta_2 v_{22}(0)}{2\omega_1^3} \frac{d^k}{dx^k} \sinh\left(s\omega_1 x\right)$$

$$+ \frac{d^k}{dx^k} \left( \frac{s^2 \delta_2 v_{22}'(0)}{2\omega_2^2} - \frac{s^2 \delta_1 v_{22}'(0)}{8} \int_x^0 p(t)dt \right) \cosh\left(s\omega_1 x\right)$$

$$+ O\left( |s|^{k+2} e^{|s\omega_1|(1-x)} \right) \tag{34}$$

**Proof.** It is similar to Lemma 2. □

Now, we are concerning to estimate upper bounds for the fundamental solutions of the problem (1)–(9), under some consideration. First of all, from Equation (28), we have

$$u_{11}(x, \lambda) = \frac{\alpha_2}{2} \left[ \cos\left(sw_1(x+1)\right) + \cosh\left(sw_1(x+1)\right) \right]$$

$$+ \frac{\alpha_1}{2(sw_1)^3} \left[ \sin\left(sw_1(x+1)\right) + \sinh\left(sw_1(x+1)\right) \right]$$

$$+ \frac{w_1}{2s^3} \int_{-1}^x \left[ \sin\left(sw_1(x-t)\right) - \sinh\left(sw_1(x-t)\right) \right] p(t) u_{11}'(t) dt$$

$$+ \frac{w_1}{2s^3} \int_{-1}^x \left[ \sin\left(sw_1(x-t)\right) - \sinh\left(sw_1(x-t)\right) \right] q(t) u_{11}(t) dt \tag{35}$$

Let $s = \sigma + i\tau$, $M = \max_{x \in [-1,0)} e^{|w_1(x+1)|}$, $Q(t) = \max_{t \in [-1,0)} \{ |p(t)| \frac{|u_{11}'(t)|}{|u_{11}(t)|}, |q(t)| \}$, $\int_{-1}^x |Q(t)| dt < \infty$ and $|\tau| \leq |\sigma|$, without loss of generality. Then the following relations $|\sin z| \leq e^{|Im\ z|}$, $|\sinh z| \leq e^{|Re\ z|}$, $|\cos z| \leq e^{|Im\ z|}$ and $|\cosh z| \leq e^{|Re\ z|}$, imply the following inequalities: $|\sin sw_1(x+1)| \leq M^{|\tau|} \leq M^{|\sigma|}$, $|\sinh sw_1(x+1)| \leq M^{|\sigma|}$, $|\cos sw_1(x+1)| \leq M^{|\tau|} \leq M^{|\sigma|}$ and $|\cosh sw_1(x+1)| \leq M^{|\sigma|}$. Hence, we obtain an upper bound for the solution (23):

$$|u_{11}| \leq M^{|\sigma|} C_1 e^{C_2}$$

where, $C_2 = \frac{|w_1|}{|s|^3} \int_{-1}^x |Q(t)| dt$ and $C_1 = \left( \alpha_2 + \frac{\alpha_1}{|sw_1|^3} \right)$. In the same way as above, we can estimate upper bounds for each of the Equations (24)–(34).

## 4. Accurate Asymptotic Behavior of the Egenvalues and Eigenfunctions

In this section, we estimate the accurate asymptotic formulas for the eigenvalues and eigenfunctions of the problem (1)–(9). According to Theorem 3, the eigenvalue problem (1)–(9) possesses infinitely many positive and negative eigenvalues. From Theorem 4, we conducted that the zeros of $W_1(\lambda)$ are the eigenvalues of the problem. First of all, we have to approximate an expression for $W_1(\lambda)$:

**Theorem 5.** *If $\lambda = s^4$, $s = \sigma + i\tau$, then $W_1(\lambda)$ (Consequently $W_2(\lambda)$), for a sufficiently large $|\lambda|$, has the following expression formula:*

$$W_1(\lambda) = \left(\frac{s^{12}\delta_1\delta_2\alpha_2\beta_2}{2(\omega_1 + \omega_2)^4}\right)\left(\sinh(s(\omega_1 + \omega_2))\sin(s(\omega_1 + \omega_2)) + \frac{(\omega_1 + \omega_2)\int_{-1}^{1}p(x)dx}{2s^2}\right.$$

$$\left. + \frac{(\omega_1 + \omega_2)\int_{-1}^{1}p(x)dx}{2s^2}\cosh(s(\omega_1 + \omega_2))\cos(s(\omega_1 + \omega_2))\right)$$

$$\times [\cosh(s\omega_2) + \cos(s\omega_2)]^2 + O\left(|s|^9 e^{4|s(\omega_1 + \omega_2)|}\right) \tag{36}$$

**Proof.** Since the functions $u_{11}(x,\lambda)$ and $u_{21}(x,\lambda)$ satisfy the initial conditions (15)–(17) and differentiating (32)–(34), then substituting the results in the characteristic equation $W_1(\lambda)$, we obtain

$$W_1(\lambda) = \begin{vmatrix} \alpha_2 & 0 & v_{11}(-1,\lambda) & v_{21}(-1,\lambda) \\ 0 & \beta_2 & v_{11}'(-1,\lambda) & v_{21}'(-1,\lambda) \\ 0 & -\beta_1 & v_{11}''(-1,\lambda) & v_{21}''(-1,\lambda) \\ -\alpha_1 & 0 & v_{11}'''(-1,\lambda) & v_{21}'''(-1,\lambda) \end{vmatrix}$$

$$= \alpha_2\beta_2 \begin{vmatrix} v_{11}''(-1,\lambda) & v_{21}''(-1,\lambda) \\ v_{11}'''(-1,\lambda) & v_{21}'''(-1,\lambda) \end{vmatrix} + O\left(|s|^9 e^{4|s(\omega_1 + \omega_2)|}\right) \tag{37}$$

By calculating $v_{i1}^{(k)}(-1,\lambda)$, for $i = 1,2$, $k = 2,3$ and substituting them into Equation (37), then we obtain (36). By similar argument, we can obtain same formula for the characteristic equation $W_2(\lambda)$. □

Note that, if $\lambda$ is a negative real number, say $\lambda = -r^2$, then we can easily show that $W_1(-r^2) \to \infty$ for a sufficiently large value of $r$. This fact, provided that the real eigenvalues of the problem (1)–(9) are bounded below, so we can say that the eigenvalues are of the form of an increasing sequence, $\lambda_0 \leq \lambda_1 \leq \dots$. In the next theorem we establish the asymptotic formula for the eigenvalues:

**Theorem 6.** *Let $\{\lambda_n\}_0^\infty = s_n^4$, be the sequence of eigenvalues of the problem (1)–(9). Then $\lambda_n$ has the following two asymptotic formulas:*

$$\sqrt[4]{\lambda_{n,1}} = \frac{1}{\sqrt{2}} \pm \frac{\pi(4n+3)}{4(\omega_1 + \omega_2)} - \frac{8(\omega_1 + \omega_2)^2 \int_{-1}^{1}p(x)dx}{(\pi(4n+3))^2} + O\left(\frac{1}{n^3}\right) \tag{38}$$

$$\sqrt[4]{\lambda_{n,2}} = -\frac{1}{\sqrt{2}} \pm \frac{\pi(4n+3)}{4(\omega_1 + \omega_2)} - \frac{8(\omega_1 + \omega_2)^2 \int_{-1}^{1}p(x)dx}{(\pi(4n+3))^2} + O\left(\frac{1}{n^3}\right) \tag{39}$$

**Proof.** Since the eigenvalues of the problem is the zeros of the characteristic equation $W_1(\lambda)$ by Theorem 4. Then from Equation (36) we have,

$$\sinh(s(\omega_1 + \omega_2))\sin(s(\omega_1 + \omega_2)) + \frac{(\omega_1 + \omega_2)\int_{-1}^{1}p(x)dx}{2s^2}\cosh(s(\omega_1 + \omega_2))$$

$$\times \cos(s(\omega_1 + \omega_2)) = -\frac{(\omega_1 + \omega_2)\int_{-1}^{1}p(x)dx}{2s^2} + O\left(|s|^9 e^{4|s(\omega_1 + \omega_2)|}\right) \tag{40}$$

Let $f(s)$ and $g(s)$ be the left and right side of Equation (40). It is easy to show that $|f(s)| \leq |g(s)|$. Then by Rouch's theorem $f(s)$ and $f(s) + g(s)$ have the same number of zeros. The zeros of the function are near to $\frac{(4n+3)}{4(\omega_1 + \omega_2)}\pi$, for a sufficiently large integer $n$, see Figure 1. Hence we obtain (38) and (39). □

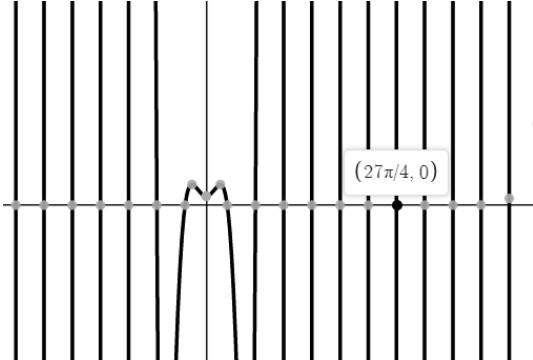

**Figure 1.** Zeros of the function $f(s)$.

According to the asymptotic expressions of the fundamental system of solutions, eigenvalues and characteristic equation $W(\lambda)$. We evaluate asymptotic formulas for the eigenfunctions. Hence, the characteristic equation $W(\lambda)$ is significant for estimating eigenvalues and eigenfunctions of the problem (1)–(9) (see [14] (Theorem 4.1)).

**Theorem 7.** *Asymptotic formulas for the eigenfunctions of the eigenvalue problem (1)–(9) corresponding to the eigenvalues $\lambda_{n,1}$, $\lambda_{n,2}$ has the following forms:*

$$\phi_{n,1}(x,\lambda_{n,1}) = \sinh\left(\frac{(4n+3)\pi}{4}x\right)\sin\left(\frac{(4n+3)\pi}{4}x\right) + O\left(\frac{1}{n}\right) \tag{41}$$

$$\phi_{n,2}(x,\lambda_{n,2}) = \frac{4}{(4n+3)\pi}\cosh\left(\frac{(4n+3)\pi}{4}x\right)\cos\left(\frac{(4n+3)\pi}{4}x\right) + O\left(\frac{1}{n}\right) \tag{42}$$

**Proof.** From Equations (23), (25), (31) and (34) we can estimate $u_{11}^k(x,\lambda_{n,1})$, $u_{21}^k(x,\lambda_{n,1})$, $v_{11}^k(x,\lambda_{n,1})$ and $v_{21}^k(x,\lambda_{n,1})$, then substituting these into the characteristic equation $W(\lambda)$. Hence, by a simple calculation, (41) and (42) can be obtained. □

Note that, from (41) and (42), we can verify that the formulas (38) and (39) are simple. In general, the eigenvalues of the problem (1)–(9) are simple (see [9] (Theorem 4.2).

**Example 1.** *If $p(x) = 0$ and $\omega(x) = 1$, then Equation (1) reduced to*

$$L[y] = y^{(4)}(x) + q(x)y(x) = \lambda y(x) \tag{43}$$

*with the conditions (2)–(9). In Reference [2], Şen investigated (43). He obtained asymptotic formulas for the fundamental solutions and eigenvalues for the problem. In this paper, we obtain an accurate asymptotic formulas for the eigenvalues for this problem. Substituting $p(x) = 0$ and $\omega(x) = 1$ into (38) and (39), then we have the following asymptotic formulas for the eigenvalues:*

$$\sqrt[4]{\lambda_{n,1}} = \frac{\pi(2n+1)}{2} + O\left(\frac{1}{n^3}\right) \tag{44}$$

$$\sqrt[4]{\lambda_{n,2}} = \frac{\pi(2n-1)}{2} + O\left(\frac{1}{n^3}\right) \tag{45}$$

*Moreover, we show that the following formulas form asymptotic formulas for the eigenfunctions for Şen's problem [2]:*

$$\phi_{n,1}(x, \lambda_{n,1}) = \sinh\left(\frac{(2n+1)\pi}{2}x\right)\sin\left(\frac{(2n+1)\pi}{2}x\right) + O\left(\frac{1}{n}\right) \tag{46}$$

$$\phi_{n,2}(x, \lambda_{n,2}) = \frac{2}{(2n-1)\pi}\cosh\left(\frac{(2n-1)\pi}{2}x\right)\cos\left(\frac{(2n-1)\pi}{4}x\right) + O\left(\frac{1}{n}\right) \tag{47}$$

**Example 2.** *We can note that in Equation (1), the weight functions $\omega_1(x)$ and $\omega_2(x)$ are constant functions on the intervals $[-1,0)$ and $(0,1]$ respectively. In this example, we consider a special non constant real valued function:*

$$\omega(x) = \begin{cases} \sqrt[4]{1-x}; & x \in [-1,0) \\ \sqrt[4]{x-1}; & x \in (0,1] \end{cases} \tag{48}$$

*Since $\omega_1(x)$ and $\omega_2(x)$ and their derivatives are bounded on the intervals $[-1,0)$ and $(0,1]$, then from Equations (37) and (40), we obtain the following approximate expression for the eigenvalues of the problem:*

$$\sqrt[4]{\lambda_{n,1}} = \frac{1}{\sqrt{2}} \pm \frac{\pi(4n+3)}{4(\sqrt[4]{1-x}+\sqrt[4]{x-1})} - \frac{8(\sqrt[4]{1-x}+\sqrt[4]{x-1})^2\int_{-1}^{1}p(x)dx}{(\pi(4n+3))^2} + O\left(\frac{1}{n^3}\right) \tag{49}$$

$$\sqrt[4]{\lambda_{n.2}} = -\frac{1}{\sqrt{2}} \pm \frac{\pi(4n+3)}{4(\sqrt[4]{1-x}+\sqrt[4]{x-1})} - \frac{8(\sqrt[4]{1-x}+\sqrt[4]{x-1})^2\int_{-1}^{1}p(x)dx}{(\pi(4n+3))^2} + O\left(\frac{1}{n^3}\right) \tag{50}$$

*It is easy to show that Equations (41) and (42) are the eigenfunction for this particular case.*

**Author Contributions:** The authors contributed equally in preparing and writing this work.

**Funding:** This manuscript receives no funding.

**Acknowledgments:** The authors would like to give their special thanks to the referees for their valuable suggestions that have improved the quality of the presentation of this paper. Also, the authors are deeply grateful to Payman Mahmood Hamaali and Wrya Karim Kadir (University of Sulaimani, College of Science, Mathematics Department) for their highly appreciated comments and helpful suggestions.

**Conflicts of Interest:** The authors declare no conflict of interest.

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
