# Peer review of "Refinement Asymptotic Formulas of Eigenvalues and Eigenfunctions of a Fourth Order Linear Differential Operator with Transmission Condition and Discontinuous Weight Function"

_symmetry, doi:10.3390/sym11081060_

Round 1

Reviewer 1 Report

This paper is significant in asymptotically approximating eigen-elements of a certain eigenvalue problem. In this context, the authors focus on a fourth-order differential equation with particular boundary and transmission conditions. They first prove that the problem admits in finitely many positive and negative eigenvalues. Then they are concerned about the asymptotic expression of the eigen-elements. I recommend this manuscript for publication in the journal after some minor improvements:

1. Some English typos should be checked carefully.

2. It obviously lacks applications concerning the question of why people should chase solving the fourth-order eigenvalue problems. A basic reference should be sufficient. Otherwise, it should be stated that the work is of theoretical interest.

3. In the introduction, the authors should point out why the assumptions on the given functions are there.

Author Response

Dear Professor ... 

The authors would like to give their special thanks to the referees for their valuable suggestions that have improved the quality of the presentation of this paper. We revised the manuscript according to the reviewers' comments and upload the revised file. In this note, we revised the manuscript according to first reviewer's comments:

We checked the article for correcting the English typos. More and more authors have been interested in investigating this kind of subject in latest years. Fourth order linear differential equations have extensive applications in different fields of engineering and science. For instance, several appropriate mathematical models have been suggested in [13,14] that assist to describe the oscillation behavior appearing in the actual suspension bridges. In the last paragraph of the introduction, we state the following sentences:

        We assume that, the solutions of (1) are satisfying the initial conditions              (15)-(19). The purpose of  these conditions is to conduct a more accurate          system of solution of the equation (1), such that satisfy the boundary and          transmission conditions.

Reviewer 2 Report

The reviewer would like to recommend this paper for publication in Symmetry. Besides, let me address some minor points to be taken into account
by the authors.

1. The authors gave an example the discontinuous weight function $\omega(x)$ in $1.10$. If it possible, please give more general examples. I am curious that the main results still holds when
\begin{equation*}
f(x)=\begin{cases}
x-1 \quad \text{if} \quad x\in [-1,0) \\
-x+1 \quad \text{if} \quad x\in [-1,0)
\end{cases}
\end{equation*}

2. Line -2 on page 4: what is the meaning of the symbol $\overline{1,3}$?

Author Response

Dear Professor ...

The authors would like to give their special thanks to the referees for their valuable suggestions that have improved the quality of the presentation of this paper. We revised the manuscript according to the reviewers' comments and upload the revised file. In this note, we revised the manuscript according to second reviewers comments:

1- We can note that in equation (\ref{eq1}) the weight function
$\omega_{1} (x)$ and $\omega_{2} (x)$ are constant functions on
the intervals $[-1,0)$ and $(0,1]$ respectively.
In this example, we consider a special non constant
real valued function: \begin{equation} \omega(x)=\begin{cases} \sqrt[4]{1-x}; &x\in[-1,0) \\ \sqrt[4]{x-1}; &x\in (0,1] \end{cases} \end{equation} Since $\omega_{1} (x)$ and $\omega_{2} (x)$ and their derivatives are
bounded on the intervals $[-1,0)$ and $(0,1]$,
then from equations (\ref{pp1}) and (\ref{RR1}),
we obtain the following approximately expression for
the eigenvalues of the problem: \begin{equation} \sqrt[4]{\lambda_{n,1}}=\frac{1}{\sqrt{2}}\pm\frac{\pi(4n+3)}
{4(\sqrt[4]{1-x}+\sqrt[4]{x-1})}-\frac{8(\sqrt[4]{1-x}+
\sqrt[4]{x-1})^{2}\int_{-1}^{1}p(x)dx}{(\pi(4n+3))^{2}}
+O\left(\frac{1}{n^3}\right) \end{equation} \begin{equation} \sqrt[4]{\lambda_{n.2}}=-\frac{1}{\sqrt{2}}\pm\frac{\pi(4n+3)}
{4(\sqrt[4]{1-x}+\sqrt[4]{x-1})}-\frac{8(\sqrt[4]{1-x}+
\sqrt[4]{x-1})^{2}\int_{-1}^{1}p(x)dx}{(\pi(4n+3))^{2}}
+O\left(\frac{1}{n^3}\right)
\end{equation} It is easy to show that equations (\ref{ff}) and (\ref{ff1}) are
the eigenfunction for this particular case. See Example 2 page 13.

2-The notation i=\overline{1,3} denotes the derivatives of the solution for i=1,2,3.

Reviewer 3 Report

The article deals with the estimation of eigenvalues of a fourth-order differential equation. The problem stated by the article can be interesting, however, at present, it lacks context. Moreover, the article is not clearly written.

P1: The authors must introduce the context in which the linear operator emerges and its potential utility.

P2 : The opening sentence is unclear. The authors must provide references.

Eq. 1 denotes both the operator and the generalized eigenvalue problem. It is unclear on what functional space the operator acts and what is omega. Why is 0 excluded from the domain of y?

The following equations are confusing. What do the subscripts denote?

The same question for Eq 1.10.

Def 1.1. What is I? What do the bars signify? Is this complex conjugation?

The author uses many subscript notations without clearly defining them. Hence is it is difficult to follow and verify the results.

Def 2.2 is confusing. Z is typically reserved for the set of integers. Why is the symbol used in this different context?

Krein spaces must be introduced also at that point (p. 4).

p.4: what is sgndelta? Is this a typo?

I can not recommend publication in the present form.

Author Response

Dear Professor ...

The authors would like to give their special thanks to the referees for their valuable suggestions that have improved the quality of the presentation of this paper. We revised the manuscript according to the reviewers' comments and upload the revised file. In this note, we revised the manuscript according to third reviewers comments:

1- P1: In the third, fourth and fifth paragraphs of the introduction page 1 and 2, we introduce the linear operator and its potential utility.

2- P1: In the first paragraph, we provide references [9,10,13,14], that they include actual applications of the fourth order linear differential equations.

3- Eq1. In the equation 1 we observed that the solution must be in L2 (I), the set of all measurable functions.

4- ω(x) is the discontinuous weight function.

5- x=0 Excluded from the interval I, because of the transmission conditions at the point x=0.

6- The subscripts refers to the fact that the solutions and the weight function ω(x) are peswise function on the interval I.

7- This facts is also hold for equation 1.10.

8- Def. 1.1. In this definition I=[-1,0)∪(0,1]. The bars indicates the complex conjugate( for example (f ) Ì… is the complex conjugate of f).

9- Def 2.2. We replace the symbole Z by K, for the krein space defined there.

10- In the last paragraph of page 3, we recall the definition of Krein space.

11- sgn δ refers to the signum function of a real number δ and it is defined in equation (13)

Round 2

Reviewer 3 Report

The manuscript improved since the first version.

On the other hand, there are still points to clarify:

I think that the way how L[y] is introduced is confusing because the authors give an eigenvalue equation, which obviously, is satisfied only for particular values of y.

Example 2: Eq 49 and 50 it is written eigenvalue, however, the formula depends on x.

The English language needs moderate corrections. 

Author Response

Dear Professor

Point 1:

     From the general definition of the eigenvalue problem, we conclude that the solution of the differential operators is a particular solution such that satisfies the differential equation and boundary conditions simultaneously.

    In our problem, the solution of equation (1) must satisfy the boundary conditions (2-5) and the transmission conditions (6-9). From Lemma 1, equation (1) has a unique solution such that satisfy the initial conditions y(a)=c_1 (λ), y^' (a)=c_2 (λ), y^'' (a)=c_2 (λ), y^''' (a)=c_3 (λ), where c_i  are arbitrary continuous function of λ∈C. In particular, we have to get a solution such that satisfy the boundary and transmission conditions, for this reason, we choose the initial conditions (15-19). For more detail about these types of the eigenvalue problem, the references [9,10,11,12] are the best resources.

Point 2:

By using the properties of the functions ω_1  and ω_2, we improved equation (49) and (50) of example 2. Since |ω_1 (x)|≤2 and |ω_2 (x)|≤1 on the intervals [-1,0)  and (0,1], respectively.

Thank you very much for your helpful comments…

Best regards
